# Evaluating Representational Similarity Measures from the Lens of Functional Correspondence

**Yiqing Bo[1], Ansh Soni[2], Sudhanshu Srivastava[1], Meenakshi Khosla[1]**
[1]University of California, San Diego, {`ybo, sus021, mkhosla`}`@ucsd.edu`
[2]University of Pennsylvania, `anshsoni@sas.upenn.edu`

## Abstract

**Neuroscience and artificial intelligence (AI) both grapple with the challenge of interpreting high-dimensional neural data. Comparative analysis of such data is essential to uncover shared mechanisms and differences between these complex systems. Despite the widespread use of representational comparisons and the ever-growing landscape of comparison methods, a critical question remains: which metrics are most suitable for these comparisons? Prior work often evaluates metrics by their ability to differentiate models with varying origins (e.g., different architectures), but an alternative—and arguably more informative—approach is to assess how well these metrics distinguish models with distinct behaviors. This is crucial as representational comparisons are frequently interpreted as indicators of functional similarity in NeuroAI. To investigate this, we examine the degree of alignment between various representational similarity measures and behavioral outcomes in a suite of different downstream data distributions and tasks. We compared eight commonly used metrics in the visual domain, including alignment-based, CCA-based, inner product kernel-based, and nearest-neighbor-based methods, using group statistics and a comprehensive set of behavioral metrics. We found that metrics like the Procrustes distance and linear Centered Kernel Alignment (CKA), which emphasize alignment in the overall shape or geometry of representations, excelled in differentiating trained from untrained models and aligning with behavioral measures, whereas metrics such as linear predictivity, commonly used in neuroscience, demonstrated only moderate alignment with behavior. These findings highlight that some widely used representational similarity metrics may not directly map onto functional behaviors or computational goals, underscoring the importance of selecting metrics that emphasize behaviorally meaningful comparisons in NeuroAI research.**

**Keywords:** Representational Similarity; Vision; Deep Neural Networks; Behavior

## Introduction

Both neuroscience and artificial intelligence (AI) confront the challenge of high-dimensional neural data, whether from neurobiological firing rates, voxel responses, or hidden layer activations in artificial networks. Comparing such high-dimensional neural data is critical for both fields, as it facilitates understanding of complex systems by revealing their underlying similarities and differences.

In neuroscience, one of the main goals is to uncover how neural activity drives behavior and to understand neural computations at an algorithmic level. Comparisons across species and between brain and model representations, particularly those of deep neural networks, have been instrumental in advancing this understanding (Yamins et al. (2014); Eickenberg et al. (2017); Güçlü & Van Gerven (2015); Cichy et al. (2016); Khaligh-Razavi & Kriegeskorte (2014); Schrimpf et al. (2018, 2020); Storrs et al. (2021); Kriegeskorte et al. (2008)). A growing interest lies in systematically altering model parameters—such as architecture, learning objectives, and training data—and comparing the resulting internal representations with neural data (Yamins & DiCarlo (2016); Doerig et al. (2023); Schrimpf et al. (2018, 2020)).

Similarly, in AI, researchers are increasingly focused on reverse-engineering neural networks by tweaking architectural components, training objectives, and data inputs to examine how these modifications impact the resulting representations. However, studying neural networks in isolation can be limiting, as interactions between the learning algorithms and structured data shape these systems in ways we do not yet fully understand. Comparative analysis of model representations offers a powerful tool to probe these networks more deeply. This endeavor is rooted in the universality hypothesis that similar phenomena can arise across different networks. Indeed, a large number of studies have provided empirical evidence licensing these universal theories (Huh et al. (2024); Kornblith et al. (2019); Bansal et al. (2021); Li et al. (2015); Roeder et al. (2021); Lenc & Vedaldi (2015)) but the extent to which diverse neural networks converge to similar representations is not well understood.

Given the growing interest in comparative analyses across neuroscience and AI, a key question arises: what are the best tools for conducting such analyses? Over the past decade, a wide variety of approaches have emerged for quantifying the representational similarity across artificial and biological neural representations (Sucholutsky et al. (2023); Klabunde et al. (2023); Williams et al. (2021)). Most of these approaches can be classified as belonging to one of four categories: representational similarity-based measures, alignment-based measures, nearest-neighbor based measures, and canonical correlation analysis-based measures (Klabunde et al. (2023)). With the wide range of available approaches for representational comparisons, researchers are tasked with selecting a suitable metric. The choice of a specific metric implicitly prioritizes certain properties of the system, as different approaches emphasize distinct invariances and are sensitive to varying as-

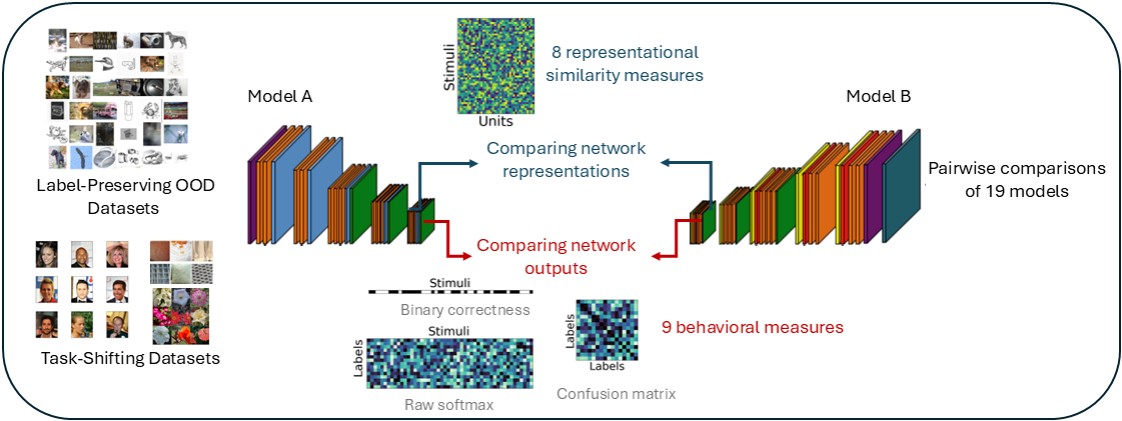

Figure 1: Framework for evaluating representational similarity metrics based on their functional correspondence. We conduct pairwise comparisons of the representational similarities and behavioral outputs of 19 vision models, utilizing 9 widely-used representational similarity measures and 10 behavioral metrics across 20 distinct behavioral datasets. **(Top Left)** Label-Preserving OOD Datasets (In-Task Distribution Shifts): Maintain ImageNet labels but alter input distributions. **(Bottom Left)** Task-Shifting Datasets (Out-of-Task Distribution Shifts): Introduce new tasks and labels, modifying both input distributions and task structures.

pects of the representations. This complexity ties into broader issues in the concept and assessment of similarity, which, as emphasized in psychology, is highly context-dependent (Tversky (1977)).

What, then, are the key desiderata for network comparison metrics? Networks may exhibit similarities in some dimensions and differences in others, but the critical question is whether these differences are functionally relevant or merely reflect differences in origin or construction. This consideration leads to a central criterion for effective metrics: behavioral differences should correspond to differences in internal representational similarity (Cao (2022)). However, identifying which measures reliably capture behaviorally meaningful differences remains an open question.

This question of whether a similarity metric captures functionally meaningful differences is particularly consequential in NeuroAI, where model–brain comparisons are frequently used to infer the computational objectives operative in the brain. For example, Yamins & DiCarlo (2016) demonstrate that "goal-driven deep learning models optimized for challenging object classification tasks produce neural response patterns that quantitatively match those observed in mid-level cortical areas," using this alignment to generate functional hypotheses about sensory processing. In the language domain, Schrimpf et al. (2021) found that models performing best on next-word prediction also excel in explaining human neural data, leading them to conclude that language-selective cortical regions are functionally tuned for predictive processing in service of meaning extraction. Complementing these findings, Richards et al. (2019) argue for a broader "deep learning framework for neuroscience" that prioritizes objective functions, learning rules, and architectures, emphasizing that systematically comparing model representations with brain data is key to uncovering the computational goals of neural circuits. In each

of these cases, the core assumption is that if a model's representation aligns with brain data under a particular metric, then the model's functional objective—such as accurate categorization or predictive processing—mirrors that of the neural circuit in question. Yet the confidence we can place in these conclusions hinges critically on whether the chosen similarity measure truly isolates behaviorally relevant structure, underscoring the importance of identifying metrics that reliably capture functionally meaningful representational differences. Our study aims to address the above challenge. Here, we make the following key contributions:

- We conduct an extensive analysis of common representational comparison measures (including alignment-based, representational similarity matrix-based, CCA-based, and nearest-neighbor-based methods) and show that these measures differ in their capacity to distinguish between models. While some measures excel at distinguishing between models from different architectural families, others are better at separating trained from untrained models.

- To assess which of these distinctions reflects differences in model behaviors, we perform complementary behavioral comparisons using a comprehensive set of behavioral metrics (both hard and soft prediction-based). We find that behavioral metrics are generally more consistent with each other than representational similarity measures.

- Finally, we cross-compare representational and behavioral similarity measures, revealing that linear CKA and Procrustes distance align most closely with behavioral evaluations, whereas metrics like linear predictivity, widely used in neuroscience, show only modest alignment. This finding offers important guidance for the selection of metric in neuroAI, where the functional relevance of representational comparisons is paramount.

**Related Work** Although few studies directly compare representational similarity measures based on their discriminative power, most efforts focus on identifying metrics that distinguish between models by their construction. These efforts typically involve assessing measures based on their ability to align corresponding layers across models with varying seeds (Kornblith et al., 2019) or identical architectures with different initializations (Han et al., 2023; Rahamim & Belinkov, 2024) or their ability to reliably separate neural responses from distinct brain areas while grouping those from the same area (Thobani et al., 2024). Closest to our work are Ding et al. (2021) and Cloos et al. (2024). The latter optimize synthetic datasets to match brain activity under different metrics, showing that even when task-relevant variables are not encoded, metrics like linear predictivity and CKA can still produce high scores. The former evaluate the sensitivity of CCA, CKA, and Procrustes to perturbations that preserve or disrupt functional behavior (e.g., seed variation, principal component deletion) in BERT (NLP) and ResNet (CIFAR-10). However, these studies examine a limited set of similarity measures and primarily assess functional similarity based on task performance alone, without evaluating the finer-grained alignment of predictions across models.

## Metrics for Representational Comparisons

**Notations and Definitions** Let $S$ be a set of $M$ fixed input stimuli. Define the kernel functions $f : S \to \mathbb{R}^{N_X}$ and $g : S \to \mathbb{R}^{N_Y}$, where $N_X$ and $N_Y$ are the output unit sizes of the first and second systems. Here, $f(s_i)$ and $g(s_i)$ map each stimulus $s_i \in S$ to vectors in $\mathbb{R}^{N_X}$ and $\mathbb{R}^{N_Y}$.

Let $X \in \mathbb{R}^{M \times N_X}$ and $Y \in \mathbb{R}^{M \times N_Y}$ be the representation matrices. For each input stimulus $s_i$, denote the $i$th row of $X$ as $\phi_i = f(s_i)$ and of $Y$ as $\psi_i = g(s_i)$, each being the activation in response to the $i$th stimulus.

**Representational Similarity Analysis (RSA)** (Kriegeskorte et al., 2008) A method that quantifies the distance between $M \times M$ Representational Dissimilarity Matrices (RDMs) of two models in response to a common set of $M$ stimuli.

$$\text{RSA}(X,Y) = \tau(\mathbf{J}_M - X^T X, \mathbf{J}_M - Y^T Y)$$

with $J_M$ denoting the $M \times M$ all-ones matrix, the representational dissimilarity matrices (RDMs) for $X$ and $Y$ are $J_M - X^T X$ and $J_M - Y^T Y$, respectively. $X^T X$ and $Y^T Y$ in $\mathbb{R}^{M \times M}$ represent the self-correlations of $X$ and $Y$, with each matrix entry $i, j$ quantifying the correlation between activations for the $i^{th}$ and $j^{th}$ stimuli. The Kendall rank correlation coefficient $\tau(\cdot)$ quantifies the similarity between these RDMs.

**Canonical Correlation Analysis (CCA)** (Hotelling, 1992) A popular linear invariant similarity measure quantifying the multivariate similarity between two sets of representations $X$ and $Y$ under a shared set of $M$ stimuli by identifying the bases in the unit space of matrix $X$ and $Y$ such that when the two matrices are projected on to these bases, their correlation is maximized.

Here, the $i^{th}$ canonical correlation coefficient $\rho_i$ (associated with the $i^{th}$ optimized canonical weights $w_x^i \in \mathbb{R}^{N_X}$ and $w_y^i \in \mathbb{R}^{N_Y}$) is being calculated by:

$$\rho_i = \max_{w_x^i, w_y^i} \text{corr}(X w_x^i, Y w_y^i)$$

subject to $\forall j < i, \quad X w_x^i \perp X w_x^j \quad \text{and} \quad Y w_y^i \perp Y w_y^j$,

with the transformed matrices $X w_x^i$ and $Y w_y^i$ being called canonical variables. To obtain a measure of similarity between neural network representations, the mean CCA correlation coefficient $\bar{\rho}$ over the first $N'$ components is reported, with $N' = \min(N_X, N_Y)$. Here,

$$\bar{\rho} = \frac{\sum_{i=1}^{N'} \rho_i}{N'} = \frac{\left\| Q_Y^T Q_X \right\|_*}{N'},$$

where $\| \cdot \|_*$ denotes the nuclear norm. Here, $Q_X = X(X^T X)^{-1/2}$ and $Q_Y = Y(Y^T Y)^{-1/2}$ represent any orthonormal bases for the columns of $X$ and $Y$.

**Linear Centered Kernel Alignment (CKA)** (Kornblith et al., 2019; Gretton et al., 2005) A representation-level comparison that measures how (in) dependent the two models' RDMs are under a shared set of $M$ stimuli. This measure possesses a weaker invariance assumption than CCA, being invariant only to orthogonal transformations, rather than all classes of invertible linear transformations, which implies the preservation of scalar products and Euclidean distances between pairs of stimuli.

$$\text{CKA}(K,L) = \frac{\text{HSIC}(K,L)}{\sqrt{\text{HSIC}(K,K)\text{HSIC}(L,L)}}$$

Here, $K$ and $L$ are kernel matrices with entries $K_{ij} = \kappa(\phi_i, \phi_j)$ and $L_{ij} = \kappa(\psi_i, \psi_j)$, where $\phi$ and $\psi$ are vectorized features from the two models. In the linear case, $\kappa$ is the inner product, so $K = X X^\top$ and $L = Y Y^\top$. HSIC evaluates the cross-covariance of the models' internal embedding spaces, focusing on the similarity of stimulus pairs.

**Mutual k-nearest neighbors** (Huh et al., 2024) A local-biased representation-level measure that quantifies the similarity between the representations of two models by assessing the average overlap of their nearest neighbor sets for corresponding features.

$$\text{MNN}(\phi_i, \psi_i) = \frac{1}{k} |S(\phi_i) \cap S(\psi_i)|$$

where $\phi_i = f(s_i)$ and $\psi_i = g(s_i)$ are features derived from model representations $f$ and $g$ given the shared stimulus $s_i$. $S(\phi_i)$ and $S(\psi_i)$ are the set of indices of the $k$-nearest neighbors of $\phi_i$ and $\psi_i$ in their respective feature spaces and $|\cdot|$ is the size of the intersection.

**Linear predictivity** An asymmetric measure of alignment between the representations of two systems, obtained using ridge regression. The numerical score is calculated by summing Pearson's correlations between each pair of predicted

and actual activations in the held-out set. For reporting, we symmetrize by averaging correlation scores from both fitting directions.

**Procrustes distance** (Ding et al., 2021; Williams et al., 2021) A rotational-invariant shape alignment distance between $X$ and $Y$'s representations after removing the components of uniform scaling and translation and applying an optimized mapping, where the mappings from one representation matrix to another is constrained to rotations and reflection. Here, the Procrustes distance is given by:

$$d(X,Y) = \min_{T \in O(n)} \|\phi(X) - \phi(Y)T\|_F$$

where $\phi(\cdot)$ denotes centering the matrix (subtracting the column-wise mean so the data is centered at the origin) and scaling it to unit Frobenius norm. i.e. $\|\phi(X)\|_F = 1$. $O(n)$ denotes the orthogonal group.

The similarity scores reported are obtained by $1 - d(X,Y)$, such that the comparison with a representation itself yields a score of 1, and lower distance yields a higher score.

**Semi-matching score** (Li et al., 2015; Khosla et al., 2024) An asymmetric correlation-based measure obtained using the average correlation after matching every neuron in $X$ to its most similar partner in $Y$. The scores reported are the average from both fitting directions.

$$s_{\text{semi}}(X,Y) = \frac{1}{N_x} \sum_{i=1}^{N_x} \max_{j \in \{1,\ldots,N_y\}} x_i^\top y_j$$

**Soft-matching distance** (Khosla & Williams, 2024) A generalization of permutation distance (Williams et al., 2021) to representations with different number of neurons. It measures alignment by relaxing the set of permutations to "soft permutations". Specifically, consider a nonnegative matrix $\mathbf{P} \in \mathbb{R}^{N_x \times N_y}$ whose rows each sum to $1/N_x$ and whose columns each sum to $1/N_y$. The set of all such matrices defines a *transportation polytope* (De Loera & Kim, 2013), denoted as $\mathrm{T}(N_x, N_y)$. Optimizing over this set of rectangular matrices results in a "soft matching" or "soft permutation" of neuron labels in the sense that every row and column of $\mathbf{P}$ may have more than one nonzero element.

$$d_{\mathrm{T}}(X,Y) = \sqrt{\min_{P \in \mathrm{T}(N_X, N_Y)} \sum_{i,j} P_{ij} \|x_i - y_j\|^2}$$

### Downstream Behavioral Measures

For classification tasks, we incorporate various downstream measurements at different levels of granularity to assess behavioral consistency across systems. For a given pair of neural networks, their activations are extracted over a shared set of stimuli. A linear readout based on a fully connected layer is trained over a training set of activations, where the resulting behavioral classification decisions determined by the linear readouts on a held-out testing set are exploited in the following ways as a comparison between the neural networks:

**Raw Softmax alignments** measure the consistency of class-level activation strengths by comparing softmax output vectors from two models. Similarity is computed as the Pearson correlation between these vectors across the test set.

**Classification Confusion Matrix alignments** measure the consistency of discrete inter-class (mis) classification patterns. A similarity score is obtained by comparing the two models' confusion matrices in the following ways:

**Pearson Correlation Coefficient** between the flattened confusion matrices given by two models, each being a vector of dimension $C^2$ over $C$ classes.

**Jensen-Shannon (JS) Distance** (Lin, 1991) introduced as a behavioral alignment measure by Tuli et al. (2021) is functionally similar to a symmetrized and smoother version of the Kullback-Leibler (KL) divergence. For class-wise JS distance, let $\hat{p} = \langle p_1, p_2, \ldots, p_C \rangle$ and $\hat{q} = \langle q_1, q_2, \ldots, q_C \rangle$ be error probability vectors over C classes, with

$$p_i = \frac{e_i}{\sum_{i=1}^{C} e_i}, \forall i \in \{1, 2, \ldots, C\}$$

where $e_i$ represents error counts per class. The JS divergence is defined as:

$$JSD(p,q) = \sqrt{\frac{D(p||m) + D(q||m)}{2}},$$

$$\text{with } D(p||m) = \sum_{i=1}^{C} p_i \log\left(\frac{p_i}{m_i}\right) \text{ and } m_i = \frac{p_i + q_i}{2}$$

A finer inter-class dissimilarity measure derived from the complete misclassification patterns shown in the non-diagonal elements of the confusion matrix results in two $C * (C-1)$ dimensional flattened vectors $\hat{p}$ and $\hat{q}$, where each component is proportional to the counts of misclassifications from class $i$ to class $j$, is calculated as

$$\frac{e_{ij}}{\sum_{i=1}^{C} \sum_{j=1, j \neq i}^{C} e_{ij}}, \quad \forall i, j \in \{1, 2, \ldots, C\}$$

The resulting distances from both methods range from [0, 1], where we simply report a similarity measure given by $1 - JSD(p,q)$.

**Classification Binary Correctness Alignments** emphasize consistency in per-stimulus prediction correctness. Error patterns for each model are encoded as binary vectors, where each entry corresponds to the correctness of a stimulus's prediction. To compare the alignment between these binary vectors, we incorporate the following measures:

**Pearson Correlation Coefficient** measures the linear correlation between the binary vectors of prediction correctness from two models, reflecting systematic agreement or disagreement in their prediction patterns over $M$ shared testing stimuli.

**Cohen's $\kappa$ Score** evaluates the agreement between two classifiers beyond chance. It is defined as:

$$\kappa_{xy} = \frac{c_{obs,xy} - c_{exp,xy}}{1 - c_{exp,xy}}$$

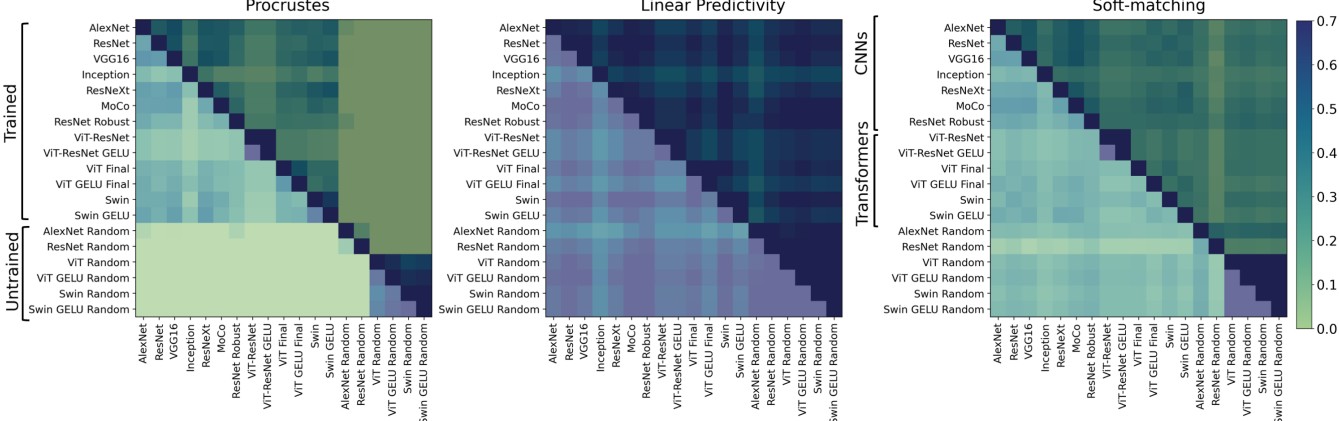

Figure 2: Model-by-model similarity matrices from different measures on the Cue Conflict task. **Left**: The Procrustes measure clearly distinguishes between trained and untrained models. **Middle**: Linear Predictivity reveals no noticeable separation between trained and untrained models or across different architectures. **Right**: Soft-matching more effectively differentiates between architectural families (CNN vs. transformers) compared to other representational metrics.

$$\text{with } c_{exp,xy} = p_i p_j + (1 - p_i)(1 - p_j)$$

where $c_{obs,xy} = \frac{\text{\# of agreements}}{M}$, $c_{exp,xy}$ is the expected probability of agreement based on the accuracies $p_x$ and $p_y$ of two independent classifiers, and $c_{obs,xy}$ is the observed probability of agreement, offering a nuanced assessment of consistency across classifications.

**Jaccard Similarity Coefficient** quantifies the agreement between two binary classifiers' predictions, defined as:

$$J(x,y) = \frac{\sum_{i=1}^{n} x_i y_i}{\sum_{i=1}^{n} (x_i + y_i - x_i y_i)}$$

where $x_i$ and $y_i$ are binary indicators of the correctness (1) or incorrectness (0) for each $i$th prediction of two models. The numerator measures the intersection, or count of samples both models predict correctly, while the denominator measures the union, accounting for samples correctly predicted by either model.

**Hamming Distance** measures the number of prediction discrepancies across a set of stimuli, defined as:

$$d(x,y) = |\{i : x_i \neq y_i, i = 1, \ldots, n\}|.$$

This metric counts the instances where the predictions from the two models differ.

**Agreement Score** quantifies the normalized difference between agreement and disagreement counts in the prediction correctness by two models:

$$s(x,y) = \frac{(n_{11} + n_{00}) - (n_{10} + n_{01})}{n_{11} + n_{00} + n_{10} + n_{01}}$$

where $i, j \in \{0, 1\}$ and $n_{ij}$ represents the number of predictions where model $x$ predicts $i$ (correct/ incorrect) and model $y$ predicts $j$, across shared stimuli.

## Downstream Behavioral Datasets

We analyze model behavior across a range of downstream tasks, spanning 20 behavioral datasets that include both in-distribution and various out-of-distribution (OOD) images and tasks (see Appendix):

**Label-Preserving OOD Datasets** retain ImageNet labels but alter input distributions to test robustness (e.g., Stylized ImageNet, ImageNet silhouettes).

**Task-Shifting Datasets** introduce new tasks and labels, requiring broader generalization (e.g., CelebA faces (Liu et al., 2015), Oxford 102 Flower (Nilsback & Zisserman, 2008)).

## Selection of Neural Network Architectures and Layers

We evaluated a diverse set of deep learning models pretrained on ImageNet-1k for 1000-class classification (Deng et al., 2009), including both convolutional networks (CNNs) and transformer-based models trained under supervised and self-supervised regimes. Architectures included AlexNet (Krizhevsky et al., 2012), ResNet (He et al., 2015), VGG16 (Simonyan & Zisserman, 2015), Inception (Szegedy et al., 2014), ResNeXt (Xie et al., 2017), MoCo (He et al., 2020), ResNet Robust (Engstrom et al., 2019), ViT-b16, ViT-ResNet, and Swin Transformer (Liu et al., 2021). Our analysis primarily targeted penultimate-layer representations, which are semantically aligned across models. For transformer models, we additionally examined final GELU activations. Randomized versions of AlexNet, ResNet, ViT, and Swin were included to assess untrained architectural behavior. See Appendix for details on our rationale for layer selection.

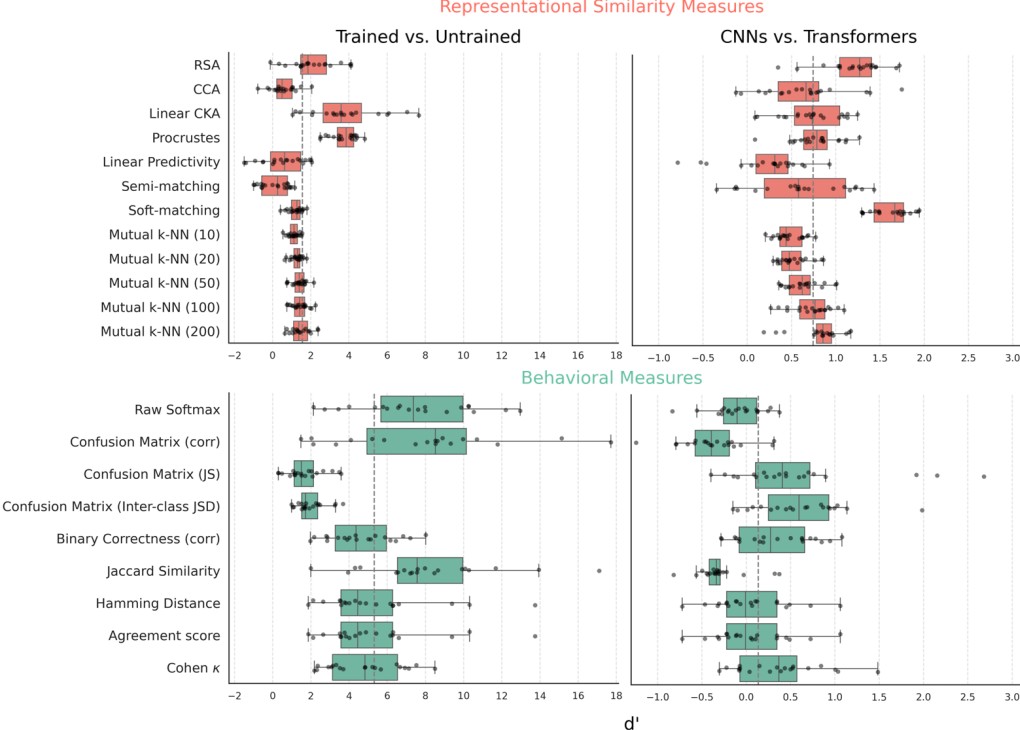

Figure 3: Discriminative ability (d' scores) of (top) representational and (bottom) behavioral similarity measures in distinguishing between trained vs. untrained models (left) and architectures (right). Each dot represents the average $d'$ score for a given dataset and metric, the dashed line indicates the overall mean $d'$ across all metrics.

# Results

## Different Representational Similarity Measures have Distinct Capacities for Model Separation

To characterize how different representational similarity measures discriminate models, we first visualize the model-by-model similarity matrices for each measure. We observed that while some measures like the soft-matching distance were effective at differentiating architectural families (Fig. 2, right), others like the Procrustes distance were more sensitive to the effects of training (Fig. 2, left), clearly separating trained from untrained models. Other measures, like linear predictivity, which allow greater flexibility in aligning the two representations, showed limited ability in distinguishing between models trained with different architectures or trained from untrained models (see Appendix for additional similarity matrices). To quantify these distinctions, we computed $d'$ scores (Appendix) to assess each measure's ability to differentiate two categories of models: (a) those from different architectural families, and (b) those with varying levels of training (trained vs. untrained). Significant differences in $d'$ scores emerged across measures (Fig. 3). For instance, Procrustes achieved $d'$ scores with a mean of 3.73 when separating trained from untrained models across all datasets, while commonly used measures like CCA and linear predictivity produced much lower scores with means of 0.57 and 0.55, respectively. Similarly, some measures were better at discriminating architec-

tural differences, with the soft-matching distance demonstrating the highest discriminability (mean of $d'$ scores = 1.61). Previous studies have also demonstrated that different measures vary in their effectiveness at establishing layer-wise correspondence across networks with the same architecture (Kornblith et al., 2019; Thobani et al., 2024). Considering these differences in how measures distinguish between models, a key question emerges: Which distinctions should we prioritize?

## Behavioral Metrics Primarily Reflect Learning Differences Over Architectural Variations

To address the question of which separation should be prioritized, we return to our central premise: measures that emphasize functional distinctions should be favored. Therefore, we next evaluated how different behavioral measures (as previously described) distinguish between models. Our results show that behavioral metrics effectively and consistently separate trained from untrained networks, with even the weakest metric (Confusion Matrix (JSD)) achieving a mean $d'$ of 1.68. However, most behavioral measures struggle to differentiate between architectural families (e.g., CNNs vs. Transformers), with the best-performing metric (Confusion Matrix (Inter-class JSD)) achieving an average $d'$ of 0.61 across all behavioral datasets (see Appendix for all similarity matrices). This suggests that differences in these architectural motifs have minimal impact on model behavior (see Appendix for further dis-

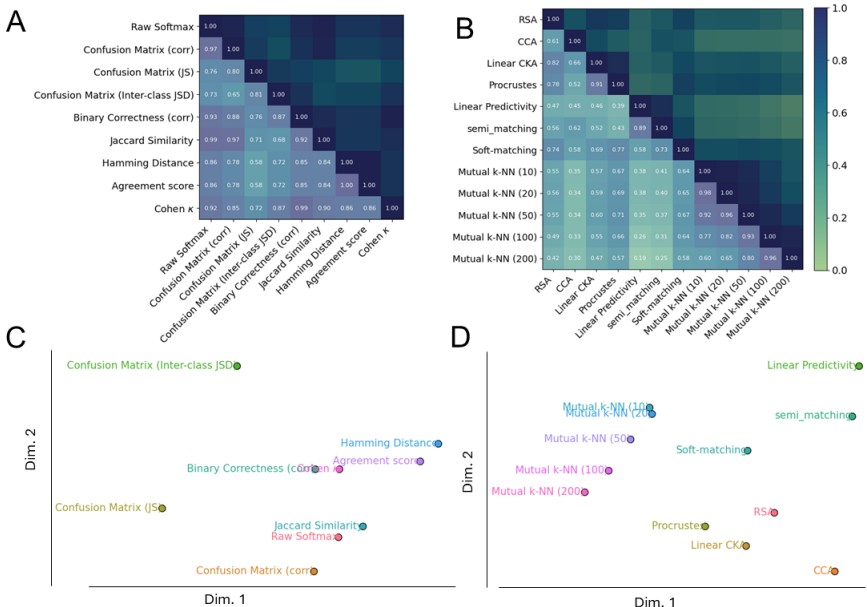

Figure 4: **Consistency Between Similarity Metrics.** (A) and (C) show the average correlation matrix and corresponding 2D multidimensional scaling (MDS) plot for behavioral similarity metrics, with distances defined as 1 minus the correlation. (B) and (D) present the same for representational similarity metrics.

cussions).

### Behavioral Metrics Show Greater Consistency Than Neural Representational Similarity Measures

We next examined the consistency across different representational similarity measures and across different behavioral measures by computing correlations between the model-by-model similarity matrices generated by each measure. As shown in Fig. 4 (Top), we find that behavioral metrics (mean r: $0.85 \pm 0.01$) are more correlated on average than representational metrics (mean r: $0.58 \pm 0.02$), with a significant difference ($z = -8.18, p = 2 \times 10^{-16} < 0.0001$).

To further understand the relationships between different representational similarity measures, we analyzed the MDS plot (Fig. 4 (Bottom)). This visualization revealed distinct clusters of measures based on their theoretical properties. Measures that rely on inner product kernels (stimulus-by-stimulus dissimilarities) tend to group together, indicating they capture similar aspects of representational structure. On the other hand, measures that use explicit, direct mappings between individual neurons—such as Linear Predictivity and Semi-Matching—form a separate cluster. Notably, Procrustes Distance and CCA also involve alignment, similar to Linear Predictivity and Semi-Matching; however, this alignment is achieved collectively across all units or neurons rather than through independently determined mappings for each neuron. Procrustes aligns the entire configuration of points, while CCA projects the two representations onto common subspaces to maximize correlation, further distinguishing them from other representational similarity approaches.

How behavioral metrics distinguish models is crucial, as most comparative analyses of representations in neuroscience and AI revolve around understanding computations and how they relate to behavior; behaviorally grounded comparisons of model representations are key to this endeavor. We find that behavioral metrics distinguish between models consistently across different datasets, reinforcing the robustness of the model relationships they uncover. The consistency of the behavioral metrics—across datasets and with each other—fulfills another scientific desideratum of replicability. Therefore, the model relationships identified by behavioral metrics are not only important but also reliable. It becomes crucial, then, to determine which representational similarity measures align with these robust behavioral relationships between models.

### Which representational similarity measures show the strongest correspondence with behavioral measures?

Given that we want to prioritize the model relationships uncovered by behavioral metrics, we move on to investigate which—if any—representational similarity metrics reveal the same underlying relationships between models. For each dataset, we computed the correlation between the model-by-model representational similarity matrix and the behavioral similarity matrix averaged across all behavioral metrics (Fig. 5). Three metrics stood out in their alignment with behavioral metrics-RSA (mean r: $0.53$), Linear CKA (mean r: $0.66$), and Procrustes (mean r: $0.70$). These same metrics also most strongly distinguished trained from untrained models (Fig. 1, top), each emphasizing global geometry or shape. In contrast, common alternatives such as linear predictivity ($r = 0.30$) and

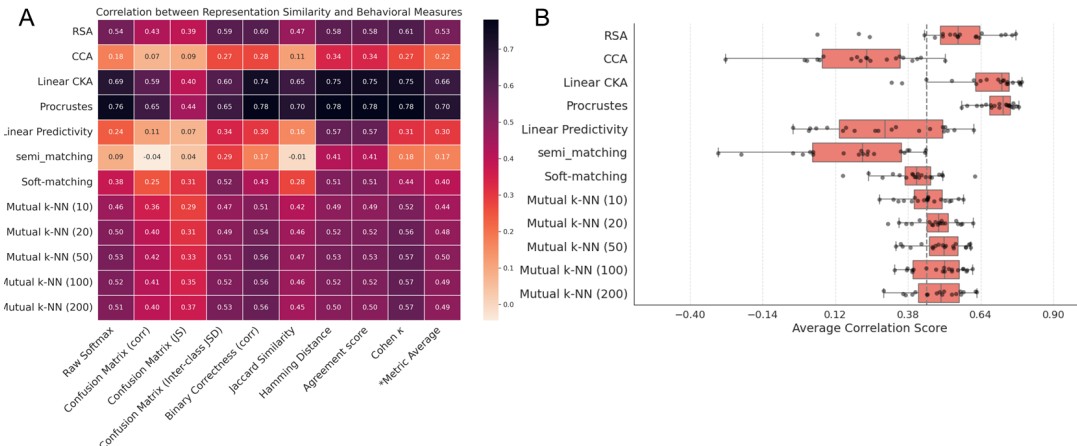

**Figure 5: Granular Comparison of Representational Similarity Measures with Behavioral Measures**. (A) Average correlation between representational and behavioral metrics across datasets. (B) Distribution of correlation scores for each representational similarity measure with behavioral measures. Each point represents the averaged score for a dataset across all behavioral measures, with the dashed vertical line indicates the overall mean correlation across all metrics.

CCA ($r = 0.22$) aligned more weakly with behavior. Given the opacity of internal representations, selecting representational similarity metrics can be challenging; these findings offer crucial guidance for metrics that support behaviorally grounded comparisons.

## Discussion

In this study, we compared 8 neural representational similarity metrics and 9 behavioral measures across 20 datasets. Based on the premise that behavioral differences should be reflected in the representational structure of neural networks, we examined how well each metric aligns with behavior. Metrics such as RSA, CKA, and Procrustes distance—which preserve the overall geometry of neural representations—tend to align closely with behavioral measures. In contrast, methods like linear predictivity, which align dimensions without preserving global geometry, show weaker alignment. This likely stems from linear predictivity's ability to map complex, distributed structures to simpler, compressed ones while still maintaining prediction accuracy. For example, trained networks can predict untrained network activations with high symmetrized scores. This overly flexible nature has also been noted in recent work (Khosla et al., 2024; Schaeffer et al., 2024). Nonetheless, linear predictivity remains valuable in applications such as BCIs, neural population control (Bashivan et al., 2018), and in-silico hypothesis generation (Jain et al., 2024).

While behavioral measures are generally consistent with one another, representational similarity metrics vary widely, underscoring the need for a deeper understanding of how these metrics discriminate between models in practice. Our analysis sets a new standard for representational similarity measures in neuroscience and AI, using downstream behavioral robustness as a guide for selecting the most suitable metric. This framework is especially crucial in model-brain comparisons, where representational analyses are frequently applied to assess if artificial neural networks and biological systems are serving comparable functional roles in terms of perceptual and cognitive processes.

Our framework for selecting representational similarity metrics, while robust, rests on certain assumptions. It presumes a particular mechanism by which behavior is read out from internal representations. Different readout strategies—especially biologically inspired ones like sparse decoding—could yield different results, particularly if some models encode behaviorally relevant information in sparser or more localized ways. In such cases, the unit-level representational structure becomes crucial.

Moreover, our evaluation focused on coarse model distinctions (e.g., trained vs. untrained, CNNs vs. transformers) and did not fully explore finer-grained variations such as subtle architectural tweaks or differences in initialization. Extending the framework to these subtler perturbations remains an important direction for future work. Further, our use of $d'$ to quantify model separability (which assumes Gaussian distributions) and Pearson correlations for behavioral vectors (which assume linearity) constitutes a statistical limitation; future work should explore alternatives such as Spearman rank correlations and other robust effect size measures.

Our desideratum involves selecting metrics which correlate with behavioral metrics to ensure that representational similarity metrics do not falsely indicate high internal similarity when behavioral differences are large. However, this does not imply that all representational similarity metrics should replicate behavioral metrics exactly. Overparameterized networks can achieve similar input–output mappings through distinct internal implementations, so a departure from a strict one-to-one correspondence between representations and behavior does not in itself detract from a metric's validity.

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

## Downstream Behavioral Datasets

**Label-Preserving OOD Datasets (In-Task Distribution Shifts)** Datasets are directly drawn from Geirhos et al. (2019); Wang et al. (2019); Geirhos et al. (2021), sharing the coarser 16 labels from ImageNet. These consist of a subset of the ImageNet1k validation set sampled from the following categories: Airplane, Bear, Bicycle, Bird, Boat, Bottle, Car, Cat, Chair, Clock, Dog, Elephant, Keyboard, Knife, Oven, Truck. These datasets maintain the original classification task but introduce distribution shifts in the input data.

- **Colour**: Served as a baseline in-distribution dataset, with half of the images randomly converted to greyscale and the rest kept in original color. Includes a total of 1280 images (80 images per label).

- **Stylized ImageNet (SIN)**: Textures from one class are applied to shapes from another while maintaining object shapes. Shape labels are used as "true labels" for confusion matrix and correctness analyses. Includes a total of 800 images

- **Sketch**: Contains cartoon-styled sketches of objects from each class, totaling 800 images.

- **Edges**: Created from the original dataset using the Canny edge extractor for edge-based representations. Includes a total of 160 images

- **Silhouette**: Black objects on a white background, generated from the original dataset. Includes a total of 160 images

- **Cue Conflict**: Images with texture conflicting with shape category, generated using iterative style transfer (Gatys et al., 2016) between Texture dataset images (style) and Original dataset images (content). Includes a total of 1280 images.

- **Contrast**: Variants of images adjusted for contrast levels. Includes a total of 1280 images.

- **High-Pass/Low-Pass**: Images filtered to emphasize either high-frequency or low-frequency components using Gaussian filters. Includes a total of 1280 images per dataset.

- **Phase-Scrambling**: Images had phase noise added to frequencies, creating different levels of distortion from 0 to 180 degrees. Includes a total of 1120 images.

- **Power-Equalisation**: Images were processed to equalize the power spectra across the dataset by setting all amplitude spectra to their mean value. Includes a total of 1120 images.

- **False-Colour**: Images had colors inverted to their opponent colors while keeping luminance constant using the DKL color space. Includes a total of 1120 images.

- **Rotation**: Images are rotated by 0, 90, 180, or 270 degrees to test rotational invariant robustness. Includes a total of 1120 images.

- **Eidolon I, II, III**: Images distorted using the Eidolon toolbox, varying coherence and reach parameters to manipulate local and global image structures. Each filtering intensity level contains 1280 images.

- **Uniform Noise**: White uniform noise added to images with a varying range to assess robustness; pixel values exceeding bounds were clipped. Includes a total of 1280 images.

**Task-Shifting Datasets (Out-of-Task Distribution Shifts)** These datasets introduce new tasks and labels, requiring models to generalize beyond the original ImageNet classification task. Each dataset consists of five trials, with non-overlapping subsets of classes selected per trial.

- **Texture** (Cimpoi et al., 2014): A texture classification dataset with 47 classes. For each trial, 9 classes are selected, each containing 120 images, resulting in 1080 images per trial (864 for training, 216 for testing). This dataset evaluates the model's capacity to classify texture patterns independent of object identity.

- **Flower** (Nilsback & Zisserman, 2008): A fine-grained classification dataset containing 102 flower species, each with a minimum of 40 images. For each trial, 20 classes are selected, yielding 800 images per trial (640 for training, 160 for testing). This dataset assesses model generalization to fine-grained natural categories.

- **Face (CelebA)** (Liu et al., 2015): A face identity classification dataset designed to test model generalization in fine-grained recognition tasks. For each trial, 50 identities are selected, each with 30 images, totaling 1500 images per trial. This dataset is of particular interest in cognitive science, given the specialized nature of face processing.

## Inter vs Intra Group Statistic Measures using $d'$ Scores

To quantify a comparative metric's ability to reflect the expected proximity between similarly trained models, compared to their dissimilarity with the untrained models, involves speculating the group statistics from the resulting similarity matrix. We employ the $d'$ score defined as:

$$d' = \frac{\mu(A) - \mu(B)}{\sqrt{\frac{\sigma_A^2 + \sigma_B^2}{2}}}$$

where $\mathbf{A}$ represents the set of similarity scores from **intra-group** comparisons, specifically the similarity scores between every pair of trained models. $\mathbf{B}$ represents the set of similarity scores from **inter-group** comparisons, specifically the similarity scores between each pair of trained and untrained

models. Equivalent to the set of entries located at the intersection of trained model rows and untrained model columns in the model-by-model similarity matrix of the metrics.

A similarity metric with $d' \geq 0$ of greater magnitude indicates a greater ability to separate trained models from untrained ones. A metric with $d' = 0$ or $d' < 0$ indicates that there were no discernible difference in average similarity scores computed in "trained model pairs" and "trained vs. untrained model pairs", or that trained vs. untrained models exhibit even higher similarity than that among trained models.

Similarly, when examining architectural differences, $\mathbf{A}$ represents intra-group comparisons within Convolutional models, while $\mathbf{B}$ captures inter-group comparisons between Convolutional models and Transformers.

## Dataset Consistency

To assess consistency across behavioral datasets, we used an $M \times M$ correlation matrix, where $M$ is the number of datasets. Each entry $i, j$ represents the correlation between datasets $i$ and $j$, derived from their downstream similarity matrices. Averaging these scores across all behavioral measures revealed high correlations, indicating consistent uniformity across most datasets.

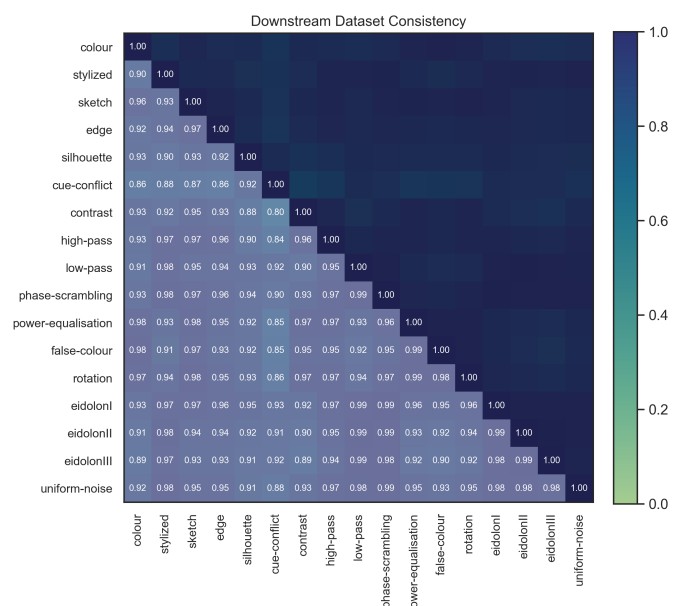

## Representation Similarity Matrices

We include the Model-by-Model Similarity Matrix given by the 8 distinct representation measures. The scores provided are averaged across 17 datasets. For mutual k-NN, different neighborhood sizes $(k)$ are included. Note that the "1 − Procrustes" score can range from $(-\infty, 1]$, whereas all other metrics yield scores within the range $[0, 1]$.

## Behavioral Similarity Matrices

Similarly, we include the Model-by-Model Similarity Matrix given by the 9 distinct behavioral measures. The scores are averaged across 17 datasets. For the measures "$1 -$ Hamming Distance" and "Agreement Scores", the alignment value can all range from $(-\infty, 1]$, whereas all other measures yield scores within the range $[0, 1]$.

## Behavioral Alignment Across Architectures

### Raw Accuracy Differences Are Present but Not Central

**Behavioral Metrics Prioritize Error Structure**  The primary goal of our behavioral evaluation is to assess similarity in error patterns — that is, whether models with comparable capabilities make similar mistakes. While transformer-based models consistently outperform CNNs in raw accuracy across both task-shift and label-preserving OOD tasks ($t = -6.41$, $p = 3.8 \times 10^{-6}$), raw accuracy alone does not capture *how* models behave. It reflects overall success rates but overlooks differences in the structure of errors. Our metrics are robust to variations in raw performance and are particularly effective for comparing models across a broad performance range. We also ensured that the datasets used exhibit sufficient variability in model accuracies, avoiding confounds due to ceiling or floor effects.

**Convergent Behavior Despite Architectural Differences**  Recent work Huang et al. (2024) suggests that, despite differing inductive biases, CNNs and ViTs often converge toward similar functional solutions during training. Consistent with this, we find that models with distinct architectures can nonetheless exhibit comparable patterns of generalization and error when trained on the same task. This convergence helps explain why architectural differences may have limited impact on behavioral error structure once task competence is achieved.

## Layer Selection and Controlled Feature Extraction

While focusing only on penultimate layers in constructing model-by-model representational similarity matrices may appear limiting, this decision was grounded in both practical and theoretical considerations. Penultimate layers provide stable, semantically aligned representations that directly influence output behavior, making them a consistent and interpretable basis for comparison across diverse architectures.

**(1) Penultimate Layers Reflect Behaviorally-Relevant Representations.** Penultimate layers directly influence model outputs and thus provide behaviorally meaningful signals. Unlike earlier layers, which capture low-level features, these final representations are more interpretable and better aligned with task-level decisions.

**(2) Intermediate Layers Are Harder to Align Across Architectures.** Comparing intermediate layers across CNNs and Transformers is challenging due to architectural differences. CNNs produce spatial maps, while ViTs output token sequences, often with a [CLS] token. In penultimate layers,

Table 1: Average correlation across datasets

| | Normalized Layer Index | | |
| | 0.6 | 0.7 | 0.8 |
|---|---|---|---|
| RSA | 0.83 | 0.80 | 0.87 |
| CCA | 0.89 | 0.89 | 0.91 |
| Linear CKA | 0.80 | 0.79 | 0.78 |
| Procrustes | 0.84 | 0.84 | 0.85 |
| Linear Predictivity | 0.84 | 0.79 | 0.85 |
| Semi-matching | 0.89 | 0.88 | 0.91 |
| Soft-matching | 0.88 | 0.88 | 0.89 |
| Mutual k-NN (10) | 0.90 | 0.89 | 0.87 |
| Mutual k-NN (20) | 0.87 | 0.87 | 0.85 |
| Mutual k-NN (50) | 0.86 | 0.85 | 0.82 |
| Mutual k-NN (100) | 0.86 | 0.85 | 0.83 |
| Mutual k-NN (200) | 0.89 | 0.86 | 0.87 |
| *Layer Mean* | $0.86 \pm 0.009$ | $0.85 \pm 0.01$ | $0.86 \pm 0.01$ |
| *Baseline Mean* | $0.74 \pm 0.02$ | $0.72 \pm 0.02$ | $0.74 \pm 0.02$ |

these differences are easier to normalize (e.g., [CLS] or average pooling for ViTs, global pooling for CNNs). Intermediate layers, however, require arbitrary choices about which token or spatial location to use, making alignment noisier.

**(3) Hierarchical Correspondence Across Layers.** To address concerns that penultimate layers may overemphasize training-related effects, we extended our analysis to include intermediate representations sampled via normalized depth indexing. Layers at normalized indices 0.6, 0.7, and 0.8 were selected to capture increasingly abstract but pre-output stages. For each model, we filtered for core representational modules (e.g., `Conv2d`, `Linear`, `attn.proj`, `mlp.fc2`) and selected the nearest valid layer to each index.

To control feature dimensionality, we applied a consistent feature extraction method: for CNNs, we used the center spatial location across all channels; for ViTs, we used the center token (excluding the [CLS] token when present). This approach avoids global pooling and keeps the features spatially specific, which is important for metrics like soft-matching.

**Results** Across all Label-Preserving OOD Datasets and multiple intermediate depths, we observed strong correlations between the representational similarity matrices of intermediate layers and those of the penultimate layer (mean $r = 0.86 \pm 0.003$), compared to a baseline correlation computed between different metrics at the same depth ($r = 0.73 \pm 0.006$). This suggests a degree of hierarchical consistency within models — that is, model-to-model representational similarity remains relatively stable across a range of layer choices (see Table 1).

## Controlling Feature Dimensionality in Soft-Matching Analyses

A potential concern is that soft-matching's sensitivity to architectural differences may stem from variation in feature dimensionality, as transformers often have larger or more variable penultimate-layer sizes (Table 2). To control for this, we re-evaluated soft-matching by randomly subsampling 768 units—the smallest shared dimensionality—across all models. Sampling was repeated over five trials and results av-

eraged. Even with equalized dimensions, soft-matching consistently distinguished CNNs from transformers on the Task-Shifting Datasets. The controlled and original similarity matrices remained highly correlated ($r = 0.997$), suggesting that the observed separation reflects genuine representational differences rather than dimensionality artifacts.

Table 2: Penultimate layer feature dimensionality across all evaluated models

| Model / Layer | Feature Size |
| --- | --- |
| AlexNet | 4096 |
| ResNet | 2048 |
| VGG16 | 4096 |
| Inception | 2048 |
| ResNeXt | 2048 |
| MoCo | 2048 |
| ResNet Robust | 2048 |
| ViT-ResNet | 768 |
| ViT-ResNet GELU | 3072 |
| ViT Final | 768 |
| ViT GELU Final | 3072 |
| Swin | 1024 |
| Swin GELU | 4096 |

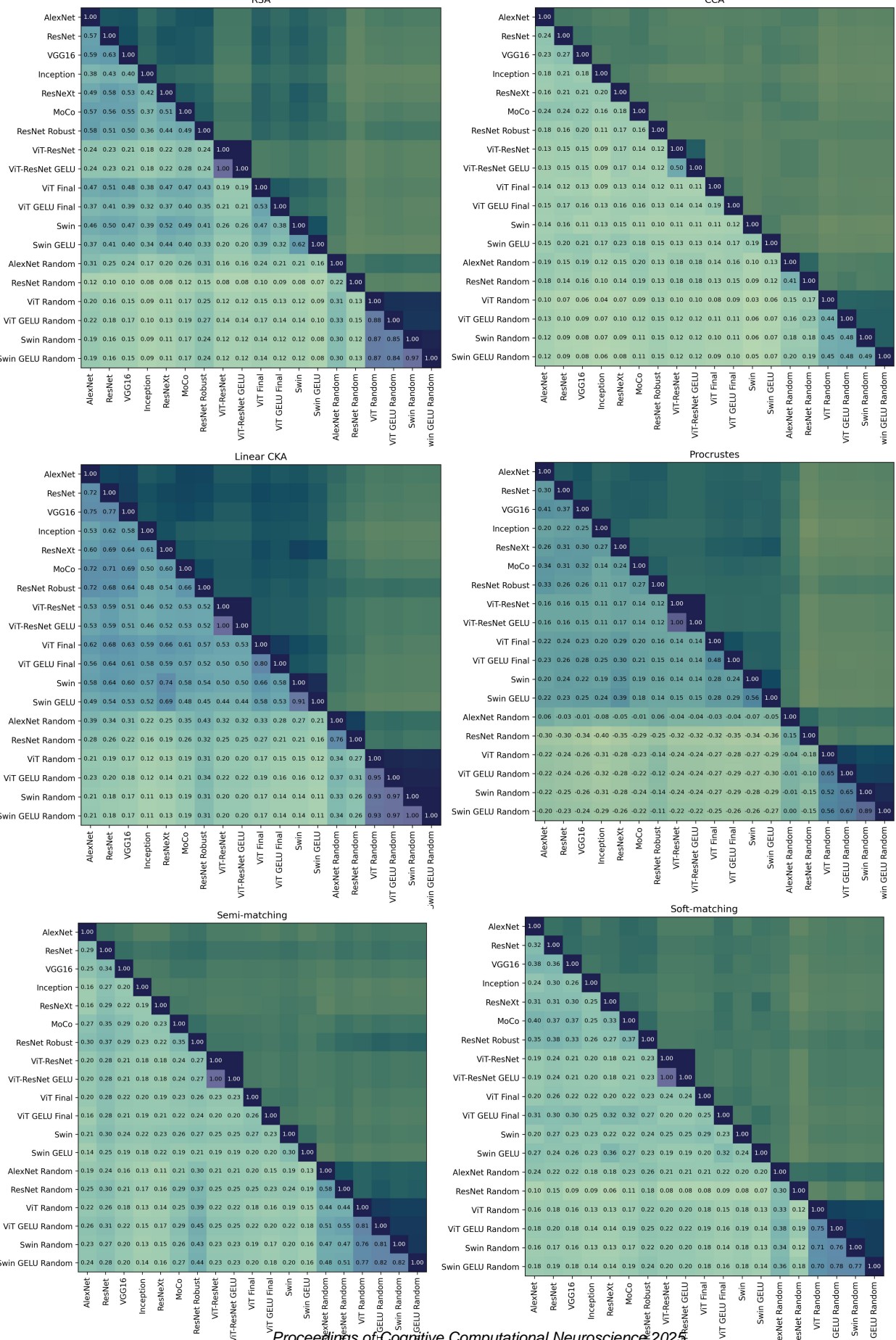

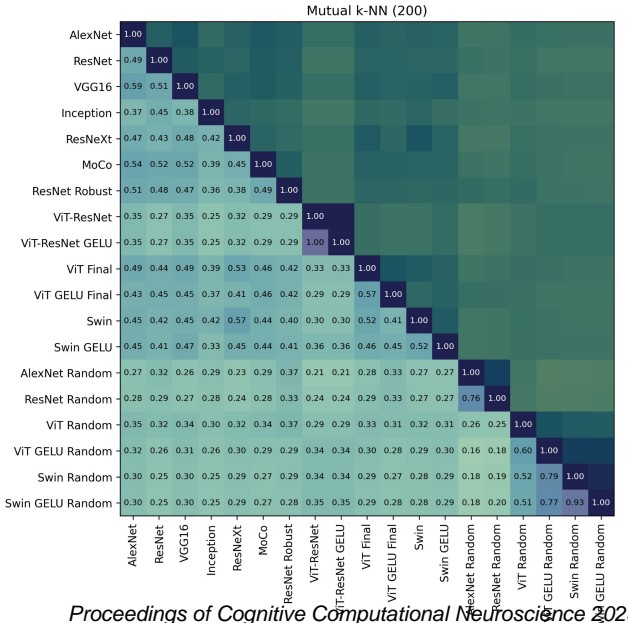

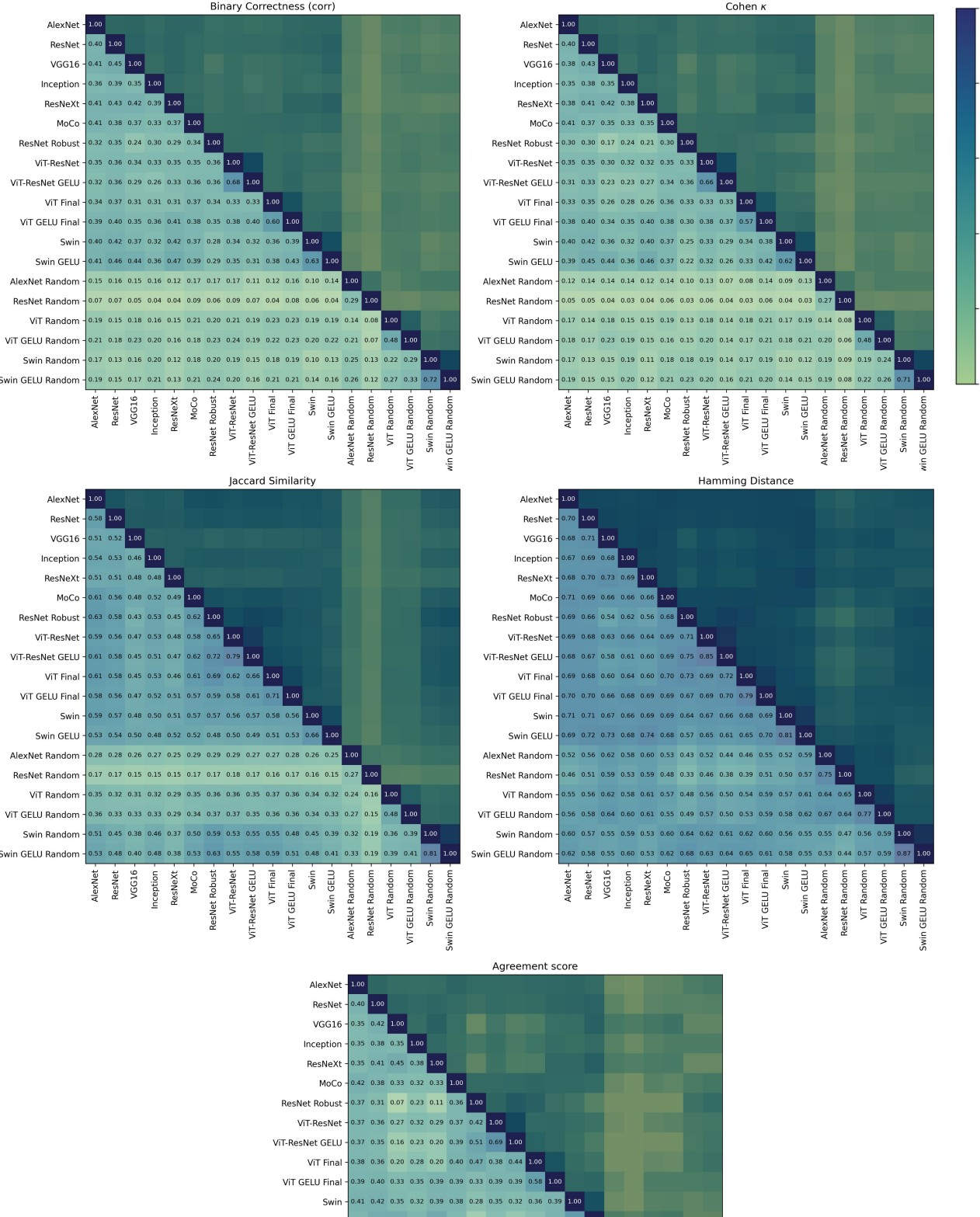

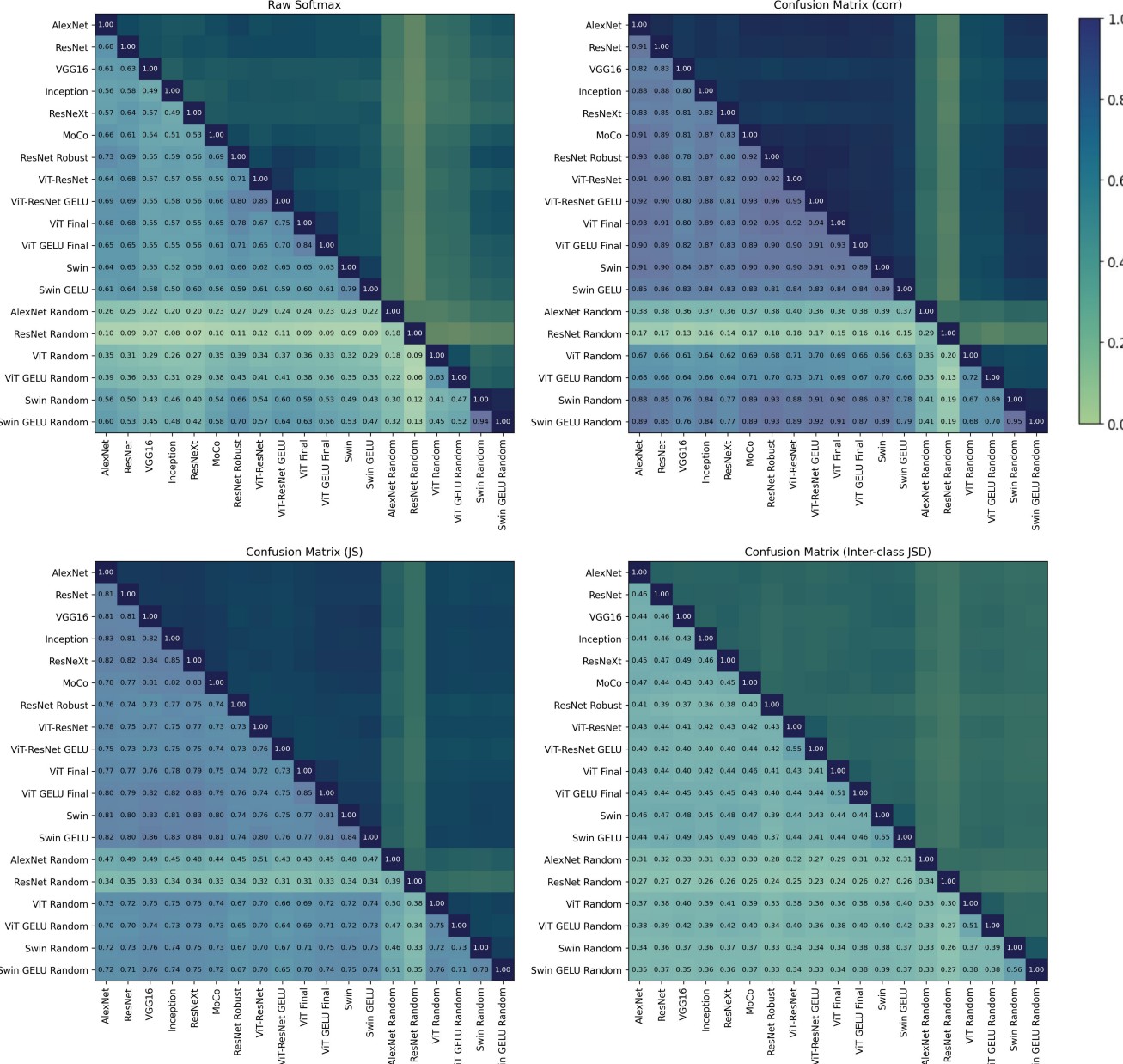

