# OpenReview forum: "Evaluating Representational Similarity Measures from the Lens of Functional Correspondence"
_ccneuro.org/CCN/2025/Proceedings — CCN 2025 Proceedings asProceedingsTalkPoster_

### Official Review · Reviewer_FyFT · 2025-03-31
**Well-framed, clearly described, systematic investigation of neural similarity metrics with convincing and actionable results**

**Soundness:** 3
**Clarity:** 3

**Comments:**

The article addresses the question: which similarly metrics are most appropriate for comparing (artificial and/or biological) neural representations? This question is highly relevant to a broad audience at CCN. The authors begin from the premise that, if ultimately we want to understand the neural mechanisms that underly behaviour (broadly construed), then we should evaluate similarity metrics by their ability to distinguish networks that behave differently. They compare a variety of neural networks using several popular similarly metrics and a variety of behavioural metrics calculated on several behavioural datasets, including tests of out-of-distribution generalization. All metrics are described clearly yet succinctly.

The authors measure model-to-model similarity using both the representational and behavioural similarity metrics. They find that different similarity measures are better for distinguishing different aspects of the models. Notably the soft-matching representational measure was best at distinguishing CNN from transformer architectures, whereas Linear CKA and Procrustes were best at distinguishing trained from untrained networks.

They find that behavioural measures are more consistent (measures are more correlated with each other) than representational similarity metrics. This supports their claim that relationships identified by behavioural metrics are reliable, in addition to being behaviourally relevant.

Finally, they look at how correlated each representational similarity metric is with each behavioural metric. Procrustes, linear CKA, and RSA stand out as considerably more correlated with the behavioural metrics. The least correlated measures are canonical correlation analysis, semi-matching, and the commonly used linear predictivity. The authors provide an intuitive intuition for why these methods fail: they are too flexible, allowing non-geometry preserving transformations when aligning representations.

This is an excellent paper whose insights will be relevant to many in our community. The didactic style and overview of metrics will be a useful resource. The results are very actionable. While others have advocated for RSA and linear CKA, the support for the Procrustes method also seems somewhat novel. I also appreciate the framing which centres the ultimate goals these metrics should serve.

Specific comments/suggestions/limitations:

* The paper includes analyses of only the penultimate network layers. This is an obvious limitation that is not discussed much. The penultimate layer might be especially appropriate for distinguishing trained from untrained networks since representations at the penultimate layer will tend to be most correlated with the output (which will obviously distinguish trained from untrained). An intermediate layer may be more appropriate for distinguishing between neural architectures as it would be least anchored to either the representations of the input or target variables.
* Is seems that, in these data, architecture is not an important determinant of behaviour. This seems somewhat counterintuitive as we know that architectural innovations have been integral for recent improvements in AI systems. How do you explain this? Is it that these networks trained on these tasks just happen to not display architecture-specific differences, but other tasks would? Or is it that your behavioural metrics are insufficient to detect these differences?
* An important contribution of this paper is to highlight the relative unsuitability of the widely used linear predictivity measure in assessing representational similarity. This point could be discussed further. Others have made a similar point on other grounds. Are there some use cases, for instance when the primary goal is accurate predictions, not understanding links between brain and behaviour, when linear predictivity remains appropriate?

Minor suggestions:

* Fig 2, Figure 5A: print x labels on an angle to save space
* ensure that all meaningful visual elements of figures are described in the figure caption. For example, I assume the dashed lines in Fig 3 and Fig 5B represent the global mean but this is not stated explicitly.

EDIT post rebuttal:

The edits proposed in response to all the reviewers will strengthen the paper. After reviewing the discussion with the other reviewers, I maintain my positive assessment of the work and high ranking on all scores.

**Expertise:**

3

**Interest:**

3

---

> ### Author Rebuttal · Authors · 2025-04-15
>
> Thank you for the constructive comments. We offer the following clarifications:
>
> 1) Focus on penultimate layer: We focused on penultimate layers to capture “readily decodable” features, and as detailed in the Appendix (“Layer Selection and Controlled Feature Extraction”), model-to-model similarity matrices are highly similar across different layer choices.
>
> 2) Lack of behavioral difference across architectures: While architectural innovations have indeed driven performance gains in AI, our findings suggest that these differences do not always translate into substantial behavioral divergences—particularly when behavior is defined in terms of error structure rather than raw accuracy.
> We observe a statistically significant accuracy advantage for transformers over CNNs across Task-Shifting and Label-Preserving OOD datasets (t = –6.41, p = 3.8e–6). However, our primary behavioral metrics emphasize the structure of model errors, which are designed to be robust to overall accuracy shifts and instead capture response similarity across stimuli.
> Moreover, recent findings (e.g., Huang et al., 2024) suggest that despite differing inductive biases, CNNs and ViTs often converge on similar functional strategies during training. This may explain why models with distinct architectures nonetheless exhibit highly aligned error patterns in our evaluations. See Appendix section on “Behavioral Alignment Across Architectures” for additional details.
>
> 3) Further discussion on linear predictivity:
> We have updated the manuscript to further clarify our use of linear predictivity. We acknowledge concerns about its flexibility, as noted in prior work (Schaeffer et al., 2024; Khosla et al., 2024). Nonetheless, linear predictivity remains a valuable metric in applications such as brain–computer interfaces, neural population control, or in cases where models are used as in-silico tools to simulate neural responses under novel conditions.
>
> 4) Adjusted figure captions.

---

> > ### Comment · Reviewer_FyFT · 2025-04-17
> >
> > Thank you for your response, which provided clarification on all of my comments. This works will guide current best practice but also inspires modifications and extensions of the approach applied here, e.g. other behavioural metrics applied on other datasets/tasks. For example, regarding point 2. I can imagine future work would want to explore behavioural metrics that are sensitive to overall shifts in accuracy, not just the structure of model errors.
> >
> > The edits proposed in response to all the reviewers will strengthen the paper. After reviewing the discussion with the other reviewers, I maintain my positive assessment of the work and high ranking on all scores.

---

> > > ### Author Response · Authors · 2025-04-21
> > >
> > > Thank you for your thoughtful feedback and support. We're glad the clarifications were helpful, and your suggestions have definitely strengthened the work and will inform future improvements.

---

### Official Review · Reviewer_UsB6 · 2025-04-01
**Well-executed, timely work; hyper-relevant to the CCN community!**

**Soundness:** 2
**Clarity:** 3

**Comments:**

**Interest**: Following the various workshops and tutorials on representational alignment metrics at last year's CCN (i.e. "Quantifying Similarity of Neural Populations" and "Battle of the Metrics"), this work strikes me a timely and deeply relevant exploration of some of the core theoretical questions the field is currently grappling with as it pertains to what we should expect to be learning from alignment metrics between neural systems. With both a comprehensive exploration of similarity metrics, and directly-linked comparisons between representational and functional / behavioral alignment, this work simultaneously and comprehensively provides direct empirical data that could well inform how similar analyses are performed in more theoretically-motivated alignment analyses in the future. This kind of "meta-benchmarking", linked directly to questions of deep conceptual and methodological interest alike (i.e. "what does the differential prediction of representation tell us about function and what metrics provide the most incisive tools for capturing the difference?") seems to me precisely the kind of work we need at the moment, and I applaud the authors' effort. The only (potential) issue I see as (potentially) limiting to more global interest are certain experimental design choices that could restrict or circumscribe the more general inferences one could make about the included alignment metrics were they to be applied in different scenarios. (More on this below).

**Soundness**: Overall, I find this work to be methodologically rich and empirically rigorous, enough I’d say it could effectively be accepted as-is with little revision. The choice of alignment metrics (from classics like regularized linear regression and RSA to new additions such as the permutation-based soft-matching metric) seems to me a well-scoped and comprehensive survey. The use of the multiple probe datasets from Geirhos et al is another strength. There are a few small statistical issues with particular combinations of behavioral metrics + accuracy measures (see minor notes below), but these can easily be fixed (or qualified with additional measures), and do not undermine the general pattern of results.

Perhaps the most substantive doubt I have in terms of “soundness” is just the question of how general we might expect the inferences from the various analyses in this work to be. As the author’s already note in the discussion, the “behavior” assessed in this work is only object categorization in trained and untrained object recognition models. The representations assessed in each model are only the penultimate layers of each.  The greater consistency of the behavioral metrics seems largely a given, considering all these models are trained on the same dataset (and architecture has often been shown to matter far less than dataset for downstream predictions of many varieties -- including categorization behavior). The authors do show that different metrics “separate” the models to greater and lesser degrees, but only of two large, pre-determined model sets: “trained” versus “untrained” models and “CNNs” versus transformers. This is definitely a good sign, but we must again consider the single-layer assessment (and should also note that the “model separation” the field tends most to be interested is “model separation” for the purpose of neural systems identification over individual models, e.g. [1])

Minor Notes:
- The use of Pearson correlation for comparisons of softmax vectors, binary correctness, and flattened confusion matrices. Pearson correlation assumes linearity, but all of these vectors (and especially the first two) violate this assumption by default. While I generally don’t fault trying out statistical metrics even when some assumptions are violated, this issue should at least be addressed, and (ideally) compared with correlation metrics more directly suited to nonlinear data (e.g. Spearman Rank order).
- Similarly, the use of d-prime in the model separability analysis assumes Gaussian-like distributions in a situation we might not expect to find them (e.g. categorization behavior).
- For measures of distributional overlap, especially in the behavioral comparisons, other effect size metrics such as the Common Language Effect Size (CLES) seem ripe for exploration.

**Clarity**: The paper is clearly written, and while the text of the figures is small, and some of the confusion matrices harder to parse than the bar plots, they are almost uniformly informative and intuitive. I would have liked to see some of the methodological specification and mathematical notation

[1] Han, Y., Poggio, T. A., & Cheung, B. (2023, July). System identification of neural systems: If we got it right, would we know?. In International Conference on Machine Learning (pp. 12430-12444). PMLR.

TLDR: **Recommendation: Strong accept!**

**Expertise:**

2

**Interest:**

2

---

> ### Author Rebuttal · Authors · 2025-04-15
>
> Thank you for the constructive comments. We offer the following clarifications:
> 1) Limited Behavioral Evaluation: We appreciate the reviewer’s concern. Although our work centers on object categorization, our behavioral evaluations extend beyond basic categorization. We also investigate broader generalization tasks—including classification of faces, flowers, and textures—and use multiple Label-Preserving OOD datasets that capture various stimulus distributions. Importantly, our findings are consistent across these different behavioral evaluations, demonstrating that our inferences about the links between different representational similarity measures and behavioral similarity generalize well within the domain of categorization behaviors. We note that classification tasks were chosen because behavioral metrics are most straightforward in these settings. Additionally, we acknowledge this limited behavioral evaluation in the discussion section of our paper. Given the time constraints and the other analyses we have included in the limited rebuttal period, this will remain a topic for future work.
>
> 2) Penultimate layers: We focused on penultimate layers to capture “readily decodable” features, and as detailed in the Appendix (“Layer Selection and Controlled Feature Extraction”), model-to-model similarity matrices are highly similar across different layer choices.
>
> 3) Model separation: Regarding model separation, our analysis reveals that even at a relatively coarse level—such as distinguishing between trained and untrained networks or between CNNs and transformers—different metrics exhibit varying sensitivities. Some metrics clearly differentiate these broad classes, whereas others do not. Determining which metrics can capture more subtle, fine-grained differences (e.g., variations among networks with the same architecture but differing in initialization or minor parameter changes) remains an intriguing open question, as noted in our discussion.
>
> 4) Use of d prime for model separability: We have highlighted the limitation in the Discussion and will explore the use of alternate model separability measures in future work.
>
> 5) Spearman Rank Correlation for Behavioral Vectors: We agree that Spearman correlation may be more appropriate. Due to time constraints, we were unable to re-run these analyses, but we acknowledge this limitation in the Discussion.

---

> > ### Comment · Reviewer_UsB6 · 2025-04-17
> >
> > Thanks to the authors' for their responses. My concerns largely remain, but I understand the limitations based on time, and will still recommend that this paper be accepted.

---

> > > ### Author Response · Authors · 2025-04-21
> > >
> > > Thank you for your careful review and support. Your suggestions and comments on statistical assumptions are appreciated and will directly inform future refinements of the analysis.

---

### Official Review · Reviewer_G1b8 · 2025-04-03
**A thorough empirical investigation into the relationship between representational similarity and function**

**Soundness:** 2
**Clarity:** 2

**Comments:**

This paper gives an important empirical analysis of mathematical techniques that are used often in this field when attempting to compare neural systems or data, but are usually undermotivated.  The authors study eight measures of representational similarity and point out how different metrics are more or less sensitive to differences in models such as architecture or trained vs. untrained.  They also study behavioral metrics, which rely on trained linear readouts from a set of activations on a classification task, which are then compared between models using various methods.  They calculate the alignment between the representational similarity metrics and the behavioral metrics, suggesting which representational similarity metrics are most sensitive to differences in what is linearly decodable from the representation. They argue that this could indicate that similarity in those metrics is more indicative of functional correspondence between the representations being compared. This is purely empirical work on deep networks (no comparisons to brain representations are made, even though this is a major motivation) and there isn't a lot of interpretation and no theoretical investigation into these ideas.  There also seem to be many factors, such as assumptions about the read out mechanism and the behavioral classification task that if changed could alter these results.  However, I still think this is important and timely work in this field, and should be discussed at the CCN meeting in some form.

I have a few questions and comments.

I am interested in more commentary from the authors about the idea that we should select a representational similarity metric based on alignment with behavioral metrics.  It is clear that behavioral metrics are useful for distinguishing models, but this seems to be a different scientific question than the one we are trying to get at when we do representational similarity, and these two clearly don't need to agree.  I can understand why large behavioral distances should correspond with measurable differences in internal representations if a metric on representations is meaningful, but is that the only desiderata?  Neural networks are highly overparametrized, and it is obviously possible for two networks to represent similar input to output functions but consist of different implementations of this function in terms of weights and activations at each layer, and these differences are what we are trying to interrogate with representational similarity measures.  This suggests to me that we should be interested in metrics that indeed do not mislead us and report that representations are similar when behavioral distance is large, but I should not necessarily expect the reverse.  That is, I wouldn't necessarily just want a representational similarity metric that replicates the behavioral metrics.  For instance in figure 5, my understanding is that you are measuring the correlation between the representational and behavior measures, so a high value here implies that the behavior and representational metrics tend to agree in both of these directions.  Do any of these representational similarity metrics tend to align with the behavioral metrics when the behavioral metric suggests dissimilarity, but less so when the behavioral metrics suggest similarity?

Congratulations on a very well-written paper.

Minor comments/questions:

- Why did you focus only on the penultimate layer of these models?

- Did you ever compare different sets of trained weights for the same models?

- Line 396, regarding the soft-matching distance--are you normalizing the representations at all for this one?  You are normalizing the representations in the Procrustes distance such that it takes a value between 0 and 1.  Are you doing the same thing for soft matching?  Is the effect that the soft-matching distance seems very sensitive to transformer vs CNN architectures explainable by very different numbers of neurons in the layers being compared between the transformers and the CNNs, or are they approximately the same size numbers of neurons in these layers?

- Line 437 - 446:  You might be interested in Harvey et. al. 2023 https://proceedings.mlr.press/v243/harvey24a which shows that Procrustes distance is also a distance on kernel matrices.

- Line 536:  You might also be interested in a paper from the same lab as above, Harvey et. al. 2024 https://arxiv.org/abs/2411.08197


Typos/grammar:

- line 387:  extra space after appendix

**Expertise:**

3

**Interest:**

2

---

> ### Author Rebuttal · Authors · 2025-04-15
>
> Thank you for the thoughtful feedback. We offer the following clarifications:
> 1. Rationale behind the proposed desideratum:
> In NeuroAI applications where representational comparisons are used to infer functional objectives of biological circuits, similarity metrics should not overlook behaviorally relevant structure. Our approach aligns with a pragmatic view of representation. As Cao (2022) argues, a representation’s value lies in its ability to support function and guide behavior. If internal differences do not track measurable behavioral changes, the metric risks lacking functional explanatory power.
> 2. Is this the only desideratum?
> Other works evaluate metrics by examining whether they can align corresponding layers across different random initializations or identical architectures with different seeds or reliably separate neural responses from distinct brain areas while grouping those from the same area  (Kornblith et al., 2019, Han et al., 2023, Thobani et al. 2024). We discuss these in the Related Work section. Each addresses distinct facets of “representational correspondence” that may not align with behavior-based metrics.
> 3. Beyond “Mirroring” Behavior: We agree that representation metrics must not always mirror behavioral metrics; our focus is on avoiding false positives where a metric claims high similarity despite marked behavioral divergence. The metrics we favor, such as Procrustes distance or linear CKA, reveal large differences in internal representations when behavioral dissimilarity is high (e.g. trained vs untrained networks).  None of the metrics automatically yield saturated maximal similarity when behavioral differences are small (e.g. as in CNNs vs transformers); rather they maintain a dynamic range suggesting that all metrics may remain sensitive to subtle differences in internal structure even when behavioral similarities are very high. We now clarify this in the Discussion.
> 4. Layer Selection: See Appendix  “Layer Selection and Controlled Feature Extraction.” Model-to-model representational similarity matrices remain highly similar across different layer choices for all metrics
> 5. Soft-Matching: We used cosine-based distances (i.e. normalized dot products), making them scale-invariant and comparable to Procrustes. We also compute softmatching after subsampling the same number of neurons from each representation and still observe CNN vs transformer representational differences (see Appendix).
> 6. Added the new citations.

---

> > ### Comment · Reviewer_G1b8 · 2025-04-17
> >
> > Thank you for your followups.  I think most of my concerns remain, in particular that this is a purely empirical survey of deep networks.  I agree with point 1.  For point 2, yes I agree these are other desiderata people use in this field to argue that a metric has meaning, but I guess I was asking you to comment on whether you think these are unreasonable and if alignment with behavioral metrics is the only desideratum.  I am satisfied with the other points except I don't see anywhere in the main text where you indicate you are using the angular distance metrics for soft-matching.  It is possible I am missing it.  I still recommend that this paper be accepted.

---

> > > ### Author Response · Authors · 2025-04-22
> > >
> > > Thank you for your continued engagement and thoughtful feedback. Regarding point 2, we note that the traditional desiderata—such as
> > > 1) matching corresponding layers across networks that differ only in random seed (Kornblith et al., 2019),
> > > 2) capturing similarity between different initializations of the same architecture (Han et al., 2023; Rahamim & Belinkov, 2024), and
> > > 3) distinguishing response patterns across brain areas while reliably grouping within-area patterns
> > > — all rest on implicit assumptions about how architectural or structural differences should manifest in representational geometry. Yet a growing body of work shows that learning can override many of these inductive biases (e.g. [1]), calling into question whether such origin‑based criteria truly reflect a metric’s functional meaning.
> > >
> > > We therefore propose that alignment with behavioral metrics be viewed not as a replacement for these established tests, but as a __complementary, functionally grounded__ desideratum—especially when one’s goal is to draw inferences about the computational objectives of a brain region based on representational comparisons. Behavioral alignment offers an agnostic criterion that directly ties representational similarity to observable performance, rather than to architectural provenance alone.
> > >
> > > Regarding soft-matching -you are correct that we did not explicitly state in the main text that our soft‑matching implementation uses angular distance. We apologize for this oversight. In the revised manuscript, we will add a clear statement that “soft‑matching distances are computed via the angular distance between component vectors."
> > >
> > > [1] Huang et al. (2024) CCN. Are ViTs as global as we think? - Assessing model locality for brain-model mapping

---

### Meta-Review · Area_Chair_NLEz · 2025-05-05

**Ccn Recommendation:** Accept as Proceedings

**Metareview:**

The reviewers converge on the view that the manuscript offers a valuable benchmark that will guide future representational-alignment studies; they agree that the methods are sound, the claims are supported by extensive evidence and the writing is good. The key reservations—limited behavioural breadth, focus on penultimate layers and some statistical-assumption mismatches—are legitimate but, in my judgement, do not undermine the central contribution: demonstrating that geometry-preserving metrics (Procrustes, CKA, RSA) align most closely with functional similarity, whereas flexible mappings (linear predictivity, CCA) are less aligned. The authors’ rebuttal satisfactorily addressed each critique, clarified implementation details and acknowledged remaining limitations in the Discussion. Given the consensus, I recommend acceptance to the Proceedings.

**Summary:**

All three reviewers find this empirical investigation of eight representational-similarity metrics and nine behavioural measures across 20 vision datasets to be timely, well-executed and clearly written. Reviewer G1b8 highlights the paper’s importance for NeuroAI, praises its didactic exposition, but worries that the study relies solely on deep-network simulations, penultimate-layer analyses and one read-out mechanism; after the rebuttal, the reviewer keeps an “adequate” rating but endorses acceptance. Reviewer UsB6 calls the work “hyper-relevant,” lauds its methodological breadth and clarity, yet notes that Pearson correlations and d′ assume linearity/Gaussianity and that the behavioural scope is confined to object categorisation; despite these caveats the reviewer issues a strong-accept recommendation. Reviewer FyFT judges interest, soundness and clarity as high, applauds the actionable outcome that Procrustes, CKA and RSA best track behaviour, but echoes concerns about layer choice and architecture-specific behaviour; the reviewer remains positive after the authors’ clarifications. Across the discussion period the authors provided reasonable point-by-point responses; none of the reviewers’ major points remained unaddressed, and all three reviewers recommended acceptance.

**Expertise:**

3